# Structure–Properties Relationship of Electrospun PVDF Fibers

**DOI:** 10.3390/nano10061221

**Published:** 2020-06-23

**Authors:** Klara Castkova, Jaroslav Kastyl, Dinara Sobola, Josef Petrus, Eva Stastna, David Riha, Pavel Tofel

**Affiliations:** 1Department of Ceramics and Polymers, Faculty of Mechanical Engineering, Brno University of Technology, Technicka 2, 616 69 Brno, Czech Republic; 2CEITEC BUT – Brno University of Technology, Purkynova 656/123, 612 00 Brno, Czech Republic; jaroslav.kastyl@ceitec.vutbr.cz (J.K.); sobola@vutbr.cz (D.S.); josef.petrus@ceitec.vutbr.cz (J.P.); eva.stastna@ceitec.vutbr.cz (E.S.); pavel.tofel@ceitec.vutbr.cz (P.T.); 3Department of Physics, Faculty of Electrical Engineering and Communication, Brno University of Technology, Technicka 10, 616 00 Brno, Czech Republic; xrihad01@stud.feec.vutbr.cz; 4Institute of Materials Science, Faculty of Chemistry, Brno University of Technology, Purkynova 112, 612 00 Brno, Czech Republic

**Keywords:** electrospinning, poly(vinylidene fluoride), surfactant, nanofiber, electroactive phase, piezoelectric activity

## Abstract

Electrospinning as a versatile technique producing nanofibers was employed to study the influence of the processing parameters and chemical and physical parameters of solutions on poly(vinylidene fluoride) (PVDF) fibers’ morphology, crystallinity, phase composition and dielectric and piezoelectric characteristics. PVDF fibrous layers with nano- and micro-sized fiber diameters were prepared by a controlled and reliable electrospinning process. The fibers with diameters from 276 nm to 1392 nm were spun at a voltage of 25 kV–50 kV from the pure PVDF solutions or in the presence of a surfactant—Hexadecyltrimethylammonium bromide (CTAB). Although the presence of the CTAB decreased the fibers’ diameter and increased the electroactive phase content, the piezoelectric performance of the PVDF material was evidently deteriorated. The maximum piezoelectric activity was achieved in the fibrous PVDF material without the use of the surfactant, when a piezoelectric charge of 33 pC N^−1^ was measured in the transversal direction on a mean fiber diameter of 649 nm. In this direction, the material showed a higher piezoelectric activity than in the longitudinal direction.

## 1. Introduction

Flexible electroactive materials play an important role in electronic applications such as sensing, actuating, energy harvesting, etc. Belonging to them, poly(vinylidene fluoride) (PVDF) has been attracting a great deal of research attention due to its outstanding pyroelectricity, piezoelectricity and ferroelectricity, as well as its relatively high dielectric permittivity [1]. PVDF’s formability, flexibility and biocompatibility further reinforce its candidature for various advanced applications [2,3,4]. Presently, PVDF-based devices come in flat films or one-dimensional (1D) fibers.

PVDF is a semi-crystalline polymer showing at least four polymorphs, of which the polar *β* phase exhibits the highest piezoelectric properties. PVDF from melt or solution crystallizes into a thermodynamically stable non-polar *α* phase. Various physical transformations like annealing, stretching or poling are carried out to transform the α phase into *β* phase.

Electrospinning as a facile and versatile technique producing nanofibers is also effective on the electroactive performance of PVDF through the *β* phase, forming directly from a solution [5,6]. In the course of the electrospinning, a high voltage is applied to the polymer solution which is ejected through a needle of a syringe. When an electrostatic force overcomes the surface tension at the tip of the needle, a so-called Taylor’s cone is formed and it is elongated into a fluid jet. The jetting fluid is collected on a collector in the form of fibers due to the potential difference between the needle tip and the collector. Besides the polymer solution’s intrinsic properties (such as polymer concentration, viscosity, solution type, surface tension, molecular weight and conformation of the polymer chain and electrical conductivity) [7,8], the electrospinning processing conditions (applied voltage, distance between needle and collector, collector’s rotational speed, solution feed rate, size of needle, material and shape of collector, etc.) [9,10,11] have a significant effect both on the *β* phase’s proportion and the crystallinity of the electrospun PVDF. Since it is known that the degree of crystallinity of PVDF can range between 35% and 70%, it is highly effective to enhance the overall PVDF crystallinity to the maximal level. The amount of crystalline phases in PVDF during a solution’s crystallization does not depend only on the temperature [12] but also significantly on the crystallization rate, which in turn depends on the solvent’s evaporation rate. Low evaporation rates favor the nucleation and growth of the thermodynamically stable *β* phase, whereas high evaporation rates yield the metastable *α* phase [13]. During the electrospinning, its own specifics contribute to the crystallization process. The possible mechanism for the formation of the *β* phase in PVDF produced by electrospinning can be explained by the high voltage applied to the polymer solution and the high stretching ratio of the fluid jets (influenced by the spinning distance and the rotational speed) [14,15]. The high ratio of stretching during electrospinning is similar to uniaxial mechanical stretching, which can cause the *β* phase transition [16]. Analogously, the low crystallization temperature, that arises from the low environmental temperature during electrospinning and thus the controlled evaporation of the solvent, can lead to the predominance of the *β* phase of PVDF [17]. The maximum fraction of the *β* phase in the range of 60–90 wt% was observed for samples with smaller fiber diameters which were electrospun at higher voltages. Decreasing the fiber diameter by up to 300 nm showed higher electrical outputs of the prepared structures [9,10]. Moreover, the poling effect of the high electrostatic field applied during electrospinning was approved in comparison to the force spinning process, where no visible piezoelectric activity was observed in the PVDF fibers containing a high fraction of *β* phase [11]. Taking account all these findings, and regarding the extremely high specificity and individuality of each piece of electrospinning equipment and each processing step, a systematic study on the relationships among the process parameters and material characteristics of the PVDF is necessary to maximize the electric behavior of the designed material.

In the present paper, a detailed study of the process and solution parameters’ influence on the fibers’ morphology, bulk density, crystallinity, phase composition and dielectric and piezoelectric characteristics is reported in order to obtain various fiber dimensions and find a clear relationship between the PVDF fibers’ structures and their electrical performance. Additionally, the optimization of the electrospinning process for preparing reproducible nano-/micro-sized fibrous PVDF layers with defined electro-activity is presented.

## 2. Materials and Methods

PVDF a with molar mass of 275,000 g mol^−1^ (Sigma Aldrich, St. Louis, MO, USA) was used for electrospinning. As solvents, dimethylsulfoxide p.a. (DMSO, Sigma Aldrich, St. Louis, MO, USA) and acetone p.a. (Ac, Sigma Aldrich, St. Louis, MO, USA) were used. CTAB (Sigma Aldrich, St. Louis, MO, USA) was added as a surfactant in given cases.

Solvents, DMSO and Ac, were mixed with a volume ratio of 7/3. The PVDF beads were dissolved in the binary solvent in a concentration of 20 wt% at 50 °C for 24 h until a visually homogeneous solution was formed. See Table 1 for a sample identification. The sample abbreviation, e.g., “50-25-15” means that the sample was electrospun at a voltage of 50 kV, in a collecting distance of 25 cm, with a collection time of 15 min on the aluminum foil, whereas the abbreviation “50-20-90-C-M” means that the sample was electrospun at a voltage of 50 kV, in a collecting distance of 20 cm, with a collection time of 90 min in the presence of CTAB on aluminum coated Mylar^®^ foil. In given cases, the CTAB surfactant in a concentration of 1 wt% (to whole mass) was added.

The prepared solutions were electrospun using the 4spin electrospinning equipment (Contipro a.s., Dolni Dobrouc, The Czech Republic), at a feeding rate of 18–20 µL min^−1^ through a needle with an inner diameter of 1.067 mm (17 G), at a collector rotational speed of 2000 rpm. The accelerating voltage was 10–50 kV. The distance between the needle tip and the collector (rotating metal drum covered by aluminum foil or Mylar^®^ foil) was 15–25 cm. The fibers were collected in the form of non-woven mats, which were characterized in this study.

The viscosity of the PVDF solutions was measured in the shear rate range of 0.1–500 s^−1^ at a temperature of 30 °C, using a rotational rheometer (HAAKE MARSII, Thermo Scientific, Dreieich, Germany) equipped with a double gap cylinder sensor system.

Measurements of the electric conductivity of the solutions were done with the conductivity meter SevenCompact S230 (Mettler-Toledo, s.r.o., Prague, The Czech Republic). Surface tension measurements of both DMSO-Ac (7/3 v/v) and PVDF solution were carried out by the tensiometer Sigma 701F (KVS Instruments Ltd., Helsinki, Finland). Due to the different viscosities of the tested samples, the Du Noüy and Wilhelmy plate methods were used for the DMSO-Ac and PVDF solutions, respectively. The parameters used for measurement were as follows: temperature 25 °C, speed up/down 20 mm min^−1^, wetting depth 6 mm, detect range 2 mN m^−1^, 10 points for each of the three measurements.

The morphology of the electrospun fibers was investigated by scanning electron microscopy (SEM, Verios 460 L, FEI Czech Republic s.r.o., Brno, The Czech Republic). The fibers’ mean diameters and 95% confidence intervals (normal distribution and level of significance, alpha = 0.05) were calculated from the SEM images. For each sample, we recorded five SEM images and fifteen diameters were measured in each SEM image (75 diameter values in total for one sample).

The bulk density (volumetric density including the spaces among fibers and the envelope volumes of the fibers themselves) of the spun PVDF layers was established by mercury intrusion porosimetry using a porosimeter, Poremaster 60 (Quantachrome, Boynton Beach, FL, USA). The bulk density $\rho_{bulk}$ was calculated based on Equation (1):(1)$$\rho_{bulk}=\frac{m_{sample}\times \rho_{Hg}}{m_{Hg}-m_{Hg+sample}+m_{sample}}$$
where $m_{sample}$ is weight of the sample, $\rho_{Hg}$ is the mercury density, $m_{Hg}$ is the weight of the mercury in a fully filled penetrometer and $m_{Hg+sample}$ is the weight of the sample and the mercury, both in the mercury filled penetrometer. The mercury density $\rho_{Hg}$ used in the calculations was 13.5387 g cm^−3^ at a temperature of 23.3 °C. Three measurements were performed for the calculation of the mean bulk density as well as the confidence intervals of Student’s probability t-distribution, at a significance level of alpha = 0.05 (95% confidence interval).

Differential scanning calorimetry (DSC, DSC Discovery, TA Instruments, New Castle, DE, USA) was performed in a temperature range of 30–200 °C, at a heating rate of 10 °C min^−1^, in the nitrogen atmosphere. The crystallinity *X_c_* was calculated according to Equation (2):(2)$$X_{c}=\frac{\Delta H_{m}}{\Delta H_{m}^{0}}\cdot 100\;(\%)$$
where Δ*H_m_* is the heat of fusion and $\Delta H_{m}^{0}$ is the heat of the fusion of 100% crystalline PVDF (104.5 J g^−1^ [18]).

The phase composition of the fibers was determined by Fourier transform infrared spectroscopy (FTIR), X-ray diffraction (XRD) and Raman spectroscopy. FTIR spectra were collected on Vertex 70 V (Bruker, Billerica, MA, USA), set up by accumulating 512 scans with a resolution of 1 cm^−1^ in transmission mode. Normalized FTIR transmission spectra after background subtraction and recalculation to absorbance were used for the quantitative evaluation of the phase ratios.

The amount of electrically active *β* and *γ* phases was calculated according to Gregorio and Cestari, as shown in Equation (3) [19]:(3)$$F\left(\beta +\gamma \right)=\frac{A_{\beta ,\gamma }}{1,26A_{\alpha }+A_{\beta ,\gamma }}$$
where *A_α_* and *A_β,γ_* are the absorbance values at 763 cm^−1^ (CH_2_ in-plane bending or rocking and CF_2_ bending and skeletal bending) and 840 cm^−1^ (CH_2_ rocking and CF_2_ asymmetrical stretching), respectively. The exclusive *β* phase peak at 1275 cm^−1^ (C–F stretching vibrations) and the *γ* phase peak at 1234 cm^−1^ (CF_2_ asymmetrical stretching and rocking) were used to determine the content of the phases separately [20].

XRD data were obtained by the X-ray powder diffractometer Rigaku SmartLab 3 kW (Rigaku Corporation, Tokyo, Japan), using a Cu Kα radiation. Raman spectra were collected by the WITec confocal Raman imaging system, alpha300 (WITec, Ulm, Germany). The excitation wavelength was 531.959 nm and the laser power was 5 mW, while the number of accumulations was 20 at an integration time of 10 sec. The AXIS SupraTM (Kratos Analytical Ltd., Manchester, UK) X-ray photoelectron spectrometer (XPS) was used to study the chemical composition of the samples. The data were collected at 15 mA of emission current with a resolution of 80 for wide spectra and 20 for element spectra.

A charge piezoelectric coefficient was measured by the electrometer 6517b (Keithley, Cleveland, OH, USA). The dielectric constant and dielectric loss were measured by a 4263B LCR Meter at a frequency of 1 kHz (HP/Agilent, Tokyo, Japan).

The piezoelectric charge coefficient *d_i_* (pC N^−1^) was evaluated by Equation (4):(4)$$d_{i}=Q/F$$
where *i* is *Tr* or *Lo* for the piezoelectric charge coefficient evaluated in the transversal or longitudinal mode (described below), respectively; *Q* is the electric charge generated by the sample and *F* is the force applied on the sample.

The electric charge was measured in the transversal mode (*d_Tr_*), as illustrated in Figure 1, and in longitudinal mode (*d_Lo_*), as illustrated in Figure 2. In the transversal mode, the sample of fibers was placed between two Cu electrodes, where the lower electrode had a diameter of d = 20 mm and the upper electrode a diameter of d = 10 mm. A mechanical force F = 0.49 N was applied on the upper electrode and the electric charge generated by the sample was measured. By this technique, each sample was evaluated in twenty positions and the resulting piezoelectric charge coefficient was represented by the average value of these obtained values.

In the longitudinal mode, the samples were mechanically stressed in the direction of the fibers, as shown in Figure 2. A PVDF fiber strap was cut from each sample with a width of 5 mm and this strap was clamped between two Cu electrodes.

Fibers of the sample were oriented in the direction of the acting force and the gap between the electrodes was 0.77 mm. One electrode was fixed and the second electrode was mechanically stressed by force F = 0.49 N. The plastic and elastic deformations of the Cu electrodes were neglected. The electric charge generated by the sample was measured during the mechanical stressing and the piezoelectric charge constant was evaluated, as mentioned in Equation (4).

## 3. Results and Discussion

### 3.1. PVDF Electrospinning Optimization

A polymer solution’s properties play a key role in the spinning process, resulting in a specific fiber morphology. Thus, in order to comprehend the process, it is essential to determine the main parts of the solution’s characteristics, namely its viscosity and conductivity. Based on our previous experiments and our knowledge of the literature [7,8,11], the fixed vol. ratio of solvents DMSO-Ac = 7/3 and the PVDF concentration in the binary mixture of 20 wt% was used for all the experiments. The CTAB, as the optimal surfactant type from the point of view of the solution preparation and the surfactant ion nature [21], was chosen. Table 2 summarizes the measured characteristics of the solvent and solutions with or without the CTAB surfactant. Since surfactants act as surface active agents, the surface tension reduction of the polymer solution was measured in the presence of the surfactant, as is evident from Table 2. The conductivity of the solvent solution was extremely low in the order of units µS cm^−1^. Nevertheless, the conductivity decreased even more after 20 wt% of PVDF was dissolved in the solvent solution, because some positively charged ions in solvents can interact strongly with fluorine atoms on the PVDF chains. It was noticeable that the addition of the surfactant significantly increased the conductivity in two orders of the units (Table 2). Moreover, the increase in conductivity caused by the ionic nature of the CTAB could bring about the intense initiation of a stable surface charge, resulting in easier drawing and stretching of the fibers. All these conditions should favor the formation of finer fibers [8,21].

In order to quantitatively compare the physical effects of CTAB on the flow properties of the PVDF solution, a rheological characterization was performed. The PVDF solutions with and without CTAB had pseudoplastic behavior (in Figure 3), i.e., with an increasing shear rate, the viscosity decreased.

The solution of DMSO-Ac in a vol. ratio of 7/3 was also characterized. The viscosity of the PVDF solutions increased by two orders of magnitude compared to the pure solution of the solvents DMSO-Ac. In Figure 3, the effect of 1 wt% of CTAB is reflected by the significant decrease in the viscosity of the PVDF solution. The shear rate in the needle was calculated from the needle’s diameter and volume flow and it was approximately equal to 0.3 s^−1^. The viscosities of all solutions are shown in Table 2 for a shear rate of 0.3 s^−1^. Thus, the better fluidity of the PVDF solution with the CTAB (i.e., lower viscosity) should reduce the resistance that the solution must overcome during spinning and the Taylor’s cone should be developed easily. With a lower viscosity, the thinner fibers can be obtained, but, on the other hand, the risk of bead formation on the fibers arises [22]. However, in our case, the viscosity was high enough that we noticed no evidence of beads on the fibers, as discussed below.

To compare the effect of the electrospinning process on PVDF’s crystallinity in further work, the PVDF solutions (without and with the CTAB) were simply cast into a petri dish in a thin layer and slowly dried at laboratory temperature. The DSC analysis showed 80.6% and 60.6% of the crystalline phase for PVDF with and without the CTAB, respectively. The crystallinity of pure PVDF is in agreement with published data referring to the crystallinity of the pure PVDF cast film from 45% to 59% with a process temperature dependence [12]. The surfactant’s incorporation into the PVDF solution resulted in fine alignment and good molecular orientation, leading to higher crystallinity, as was also shown by other authors [23].

The spinnability of the prepared pure PVDF solutions in the mixed DMSO-Ac solvent was studied at the voltages of 10 kV–25 kV–35 kV–45 kV–50 kV and at the collecting distances of 15 cm–20 cm–25 cm, as shown in Table 3.

At the lowest voltage of 10 kV, the PVDF solution was unable to spin. The lowest voltage was not enough to pull the fibers and no fibers were collected. The higher voltages (25 kV–50 kV) were already enough for fiber formation. The lowest collecting distance of 15 cm at 25 kV did not lead to stable electrospinning; the solution visibly overcame the surface forces, but the distance was probably not high enough to form a continual fibrous structure and almost drops or imperfect fibers were deposited on the collector. Exceeding the conditions of 25 kV and 15 cm (to the higher distances and voltages), the smoothest fibers without any defects or crucial imperfections were collected at differing diameter values (see Table 3 and selected samples in Figure 4).

Taking into account the benefit of a fiber alignment for potential applications [11,24], the highest collecting rate was applied in our experiments. The highly oriented arrangement of the fibers can be observed in Figure 4a, where, at an applied voltage of 25 kV, the stretched and aligned fibers were formed. In contrast, the higher voltage of 50 kV led to partially random collecting of the fibers (see Figure 4b). This phenomenon can be explained by the excessive stretching force at the ultrahigh voltage, which destabilized Taylor’s cone formation, and the aligned chains were disturbed and re-entangled [9]. The mean diameters of the fibers ranged from ca 600 nm to 1400 nm. The mean diameter as well as its confidence interval decreased with an increase in the applied voltage (see Table 3). The effect of increasing the voltage on the narrowing diameter size distribution is evident also from Figure 5. There was no other significant dependence of the collecting distance on the fiber diameters, at applied voltages of 35 kV and higher, in the whole collecting distance range, as can be seen in Figure 5.

Certain spinning instability was revealed at the relatively high voltage of 45 kV; especially when measured at the low distances of 15 cm and 20 cm, a layer of highly randomly orienteered under micrometer-size fibers was produced. The inability to align the fibers caused an increase in the fibers’ diameters. This could be the result of a decrease in the electric field’s intensity. An electric field’s intensity is not stable with a differing spinning distance at a constant voltage, so the one shorter than critical length could lead to insufficient fiber stretching, resulting in thick non-aligned fibers even at the high rotational speed of the collector [10].

The effect of the applied voltages and spinning distances on the fibers’ phase composition was further demonstrated by the determination of electroactive phases using FTIR analysis. The electroactive *β*- plus *γ*-phase content is shown in Table 3, and it was in the range of ca 82–92%. A correlation between fiber diameter and electroactive phase content is evident. The lower the fiber’s diameter, the higher the content of the electroactive phases that was obtained. The fibers with the highest values (90–92%) of electroactive phases contained also the highest amount of *β* phase itself. The *β* phase values of around 86–89% are among the highest published ones [9,10,24]. The applied voltage had a significantly higher effect on the phase composition in contrast to the stretch power, causing the high alignment. The process parameters used for the electrospinning of the thinnest fibers with the highest volume of *β* phase were stated as the optimal parameters from the point of view of a prospective composition, and these were applied in further experiments which targeted the preparation of PVDF fibrous layers that are as electro-effective as possible.

### 3.2. Characterization of Electrospun PVDF Fibers under Optimized Processing Parameters

In order to describe the relationship between the morphology and electro-behavior, the most diverging fibers with the highest and the lowest diameters, processed at 25 kV and 50 kV in the collecting distance of 20 cm, were selected for further detailed analysis of the structure and the phase composition. These samples were compared to the fibers prepared by an advanced process, including the optimal process parameters (50 kV, 20 cm), the addition of the CTAB into the PVDF solution (50-20-90-C) and the collection, with Mylar^®^ foil as the collector surface (instead of the aluminum foil) (50-20-90-C-M). Mylar^®^ foil is a good choice to substitute the conventionally used aluminum foil during electrospinning collection, as Mylar^®^ foil is conductive and stiff enough to prevent PVDF fiber shrinkage during solvent evaporation and drying, thus stabilizing an amount of the electroactive phase.

For the detailed material and functional characterization of the selected fibers, establishing the appropriate thickness of the fiber layer was necessary; hence, the collection time was increased to 90 min. The process parameters and fiber characteristics are given in Table 4.

Analysis of the fiber diameters showed conformity of the obtained results in the different collection times of 15 min and 90 min, at the same collecting distance of 20 cm. The mean diameter of the fibers spun at 25 kV was ca 1343 nm (sample 25-20-15) and 1346 nm (sample 25-20-90) after 15 and 90 min of collection time, respectively. At 50 kV, the diameter was 678 nm (sample 50-20-15) and 649 nm (sample 50-20-90) after 15 and 90 min of collection time, respectively. The fiber morphology can be seen in Figure 6a,b. The same conformity was determined for the phase content of these fibers (see Table 4). This comparison shows the time-dependent stability of the process from the point of view of fiber diameter, as well as a presumption of the morphology repeatability during the electrospinning process.

The morphology of the highly aligned fibers prepared from the PVDF solution with the CTAB (sample 50-20-90-C and 50-20-90-C-M) can be seen in Figure 6c,d. The mean fiber diameters of 474 nm and 276 nm are significantly lower, with a narrow diameter size distribution than the diameters of the fibers prepared from the solutions without CTAB. The effect of decreasing surface tension and viscosity, and increasing conductivity, on the decrease in the fiber diameter as well as on the confidence interval is evident. Moreover, the Mylar^®^ foil enhanced the forming and collection of the thinner fibers.

The overall crystallinity X_c_ increased with the increasing diameter from ca 55% to 62%, which is slightly lower than the values for the cast PVDF film with or without the CTAB. The CTAB’s presence increased the crystallinity regardless of the processing method. This meant that it was not only a reduction in the surface tension and solution conductivity during electrospinning that were the consequences that contributed to the increase in crystallinity. The hydrophobic part of the CTAB can join to the hydro-carbonic structure of the polymer chain but the hydrophilic/polar one leaves freely. The accumulation of polar groups in the polymer space resulted in repulsion between the polymer chains, and this consequently increased the chains’ organization and the PVDF’s crystallinity during electrospinning as well as during casting. The presence of CTAB in the electrospun PVDF fibers was confirmed by XPS analysis. Wide spectra (see Figure 7a) show peaks of core levels of carbon (C1s), fluorine (F1s) and oxygen (O1s). XPS spectra of the F1s’ core levels (see Figure 7b) have a similar shape to the F-C-F bond at 687.4 eV for all samples. The C1s’ spectra (see Figure 7c) confirm the presence of carbon bonds with fluorine and with hydrogen. Deconvolution of the O1s’ peaks shows that samples prepared with CTAB are enriched with oxygen containing functional groups (see Figure 7d–g).

For the detailed phase composition, the samples were comprehensively characterized by FTIR spectroscopy, XRD and Raman spectroscopy. The FTIR characteristic absorption bands that were used for the phase quantity calculation are shown in the spectra in Figure 8.

Table 4 shows the *β* phase content from 74.4 to 89.5 wt% and confirms the predominance of the *β* phase over the α and *γ* phases at lower diameters of the fibers. The *γ* phase content in the range 1.3 wt%–9.8 wt% decreased with a decrease in the fiber diameter. XRD measurements correlate with the calculations from FTIR and show dominant reflections of the *β* phase at 20.6° (110/200) and 36.6° (101) [25] (see Figure 9a). The diffraction bands of the monoclinic *α* phase peaks are located at 18.4° (020) and 41.1° (111). Since the diffraction bands are overlapped, the Lorentzian function was used to distinguish the peaks reflecting the *γ* phase at 20.3° (101) [20]. The results of peak fitting after background correction are shown in Figure 9b.

Complementary information is given by the Raman spectra (Figure 10). The relative peak intensity of the *β* phase at 839 cm^−1^ increases in comparison with the *α* phase peak at 794 cm^−1^. A decrease in the peak intensity of the amorphous halo is observed with an increase in the fibers’ crystallinity. The Raman spectra provide additional data about the skeleton chain, but the *γ* phase peak is hardly distinguished. It is situated at 811 cm^−1^ and 839 cm^−1^, where it overlaps with the *α* and *β* phase peaks.

The above described comprehensive phase composition and morphology analysis data clearly showed and confirmed the effectiveness of the electrospinning process in enhancing the PVDF’s crystallinity and the electroactive phase amounts.

The dielectric properties of PVDF polymer are based on the crystalline phase, orientation and temperature [26,27,28,29]. In a fully dense PVDF sample, the dielectric constant is approximately ε_r_ 14 for the *α* phase, and the higher content of the *β* phase can slightly decrease this value by one or more [27,28]. But the more significant effect of lowering the dielectric constant has the porosity. The higher porosity inside the material introduces a lower dielectric constant. The electrospinning procedure fabricates a fiber material with a very high porosity. The porosity can reach more than 70% [30,31]. Figure 11 shows the electrospun PVDF fibers characterized from a dielectric point of view. From our results, the increase in the dielectric constant with an increase in the size of the fiber diameter in the electrospun material is observed. This can be assigned to the decreasing content of the *β* phase and the crystallinity in samples with a higher diameter of the fibers (see Table 4). The smaller crystalline size, the presence of defects and irregularities in the samples with a lower crystallinity caused the higher mobility of the dipole moments and the growth of the dielectric constant [26,27,28,29]. The increase is not linear and the highest increase of the dielectric constant appears between the sample with the lowest size of the mean fiber diameter, i.e., 276 nm, and the second lowest size of the mean fiber diameter, i.e., 474 nm, where the dielectric constant was 1.1 and 1.7, respectively. The dielectric constant increased by nearly two times when the size of the fiber diameter was increased by approximately 200 nm and the fiber diameter was almost doubled. A further increase in the fiber size on the mean diameters of 649 nm and 1346 nm resulted in the increase of the dielectric constant to 2.0 and 2.2, respectively. This increase is not so evident for larger fiber diameters compared to the values that were observed for the lowest diameters.

The dielectric loss of the samples 25-20-90, 50-20-90, 50-20-90-C and 50-20-90-C-M nanofibers were 2.6 × 10^−3^, 2.5 × 10^−3^, 14.0 × 10^−3^ and 14.0 × 10^−3^, respectively, at 1 kHz. The increase in dielectric loss can be attributed to the larger interface of PVDF-air and the interface of nanofibers among themselves. This type of energy dissipation has been described by several authors [32,33]. Moreover, the steep difference in the dielectric loss between fibers with diameters of up to 470 nm and over 649 nm can be affected by the presence of the hygroscopic surfactant CTAB in the PVDF fibers with lower diameters. The retained irremovable water in the fibers led to hydrogen bonding among the fluorine atoms of the PVDF and the water molecules (the presence of hydrogen bonds was confirmed by XPS analysis, as discussed above) and thus the dielectric loss increased in the material [34,35].

The piezoelectric charge coefficient d_Tr_ obtained at the samples, measured in the transversal mode, is shown in Figure 12. In the figure, the increase in the piezoelectric charge coefficient value with a decrease in the diameter of the pure PVDF fibers (25-20-90 and 50-20-90) can be observed. The highest value of the d_Tr_ = 33 pC N^−1^ was observed for the sample 50-20-90, where the mean fiber diameter was 649 nm and contained the highest amounts of the electroactive phases, including the highest amount of γ phase. Then, the piezoelectric charge coefficient decreases in the value 12.3 pC N^−1^ for the sample 50-20-90-C, represented by the mean fiber diameter of 474 nm, and it decreased further in the value 5.1 pC N^−1^ for the sample 50-20-90-C-M, with the mean fiber diameter of 276 nm, as is described in Table 4. Despite the high content of β phases in both samples, the presence of CTAB in the fibers with mean diameters of 474 nm and 276 nm significantly contributed to the decrease in the piezoelectric activity due to its conductivity [36] and water retention [37].

The values d_Lo_ obtained in the longitudinal direction were lower than the values d_Tr_ measured in the transversal direction, as shown in Figure 12. The increasing trend of the piezoelectric charge coefficient in the longitudinal direction, d_Lo_, with the decrease in the fiber diameters of the pure PVDF samples is obvious and can be seen in Figure 12. The highest value d_Lo_ = 10.4 pC N^−1^ was observed for the sample 50-20-90, with the mean fiber diameter of 649 nm. Then, d_Lo_ decreased for both samples with the CTAB with a decrease in the fiber diameter, and the lowest value was obtained for the sample 50-20-90-C-M, with the lowest mean fiber diameter of 276 nm, where the d_Lo_ was around 1 pC N^−1^. The deterioration effect of CTAB on the piezoelectric activity is obvious. These determined characteristics correlate with the bulk densities of the samples (calculated according to Equation (1)), as shown in Figure 12 and Table 4. The bulk density of the spun PVDF’s fibrous layers increased with the diameter decrease in the pure PVDF fiber and, inversely, the bulk density decreased in the case of the PVDF fibers with the CTAB. The pore-forming effect in the presence of the hydrophilic surfactants (like CTAB) in polymers during their processing was reported [38,39], and the higher porosity could contribute to the deterioration of the dielectric properties of the PVDF material which is described above. Despite the assumption of a denser PVDF structure formation due to thinner and more aligned fibers, the CTAB could negatively affect the degree of porosity from the viewpoint of the piezoelectric properties as well.

As evidenced by the piezoelectric charge coefficient trends, the electrospun fibers are more piezo-electrically sensitive in the transversal direction than in the longitudinal one. This is due to the dominant polarization of the dipoles in the direction normal to the fibers’ longitudinal axis, which is connected to the potential on the collector. These findings point to the application of the fibrous material in the field of pressure active sensors (heart rate sensors in wearable textiles, precise low-pressure acoustic sensors, etc.).

From our results, it can be seen that there is an increasing trend of piezoelectric activity with the decreasing size of the fiber diameter/increasing of the *β* phase in the case of pure PVDF fibers, in accordance with published results [9,10]. Interestingly, a negative impact of the use of surfactants on PVDF’s electrical performance was found. The negative impact of CTAB in the PVDF solution exceeded its positive effect on the morphology and phase composition during the electrospinning process, which was aimed primarily at piezoelectric activity enhancing. To the best of our knowledge, the relationship between surfactant enhanced fiber diameter and fibers’ electric performance has still not been published. Our results showed the possible consequences of the chemical and physical nature of the surfactant used, and thus its selection must take into account not only its effect on spinnability but also the effect on the functionality of the resulting material. This phenomenon will be studied deeper in further work focused on the role of different surfactants on the PVDF fibers’ piezoelectric response.

## 4. Conclusions

Poly(vinylidene fluoride) (PVDF) fibrous layers with nano- and micro-sized fiber diameters were prepared by a controlled electrospinning process. The fibers with mean diameters from 276 nm to 1392 nm were spun at voltages of 25 kV–50 kV, either from the pure PVDF solutions or in the presence of the CTAB surfactant. The mean fiber diameters decreased and the *β* phase content increased with the increase in the applied spinning voltage. The presence of the CTAB surfactant enhanced the PVDF’s spinnability and crystallinity. A maximum crystallinity of 64% with 89 wt% of the *β* phase was achieved in the PVDF sample with a mean fiber diameter of 276 nm, which was spun from a solution of PVDF with 1 wt% of the CTAB surfactant. The piezoelectric activity increased with the decreasing diameter of the pure PVDF fibers. The highest piezoelectric charge of 33 pC N^−1^ was measured in the transversal direction on the sample with the mean fiber diameter of 649 nm, spun from the PVDF solution without the CTAB, containing the lowest amount of *β* phase and with the presence of the *γ* phase. Although the presence of the CTAB decreased the fibers’ diameter and increased the electroactive phase content, it evidently deteriorated the electro performance of the PVDF material. This finding is contrary to the results found in other publications, namely that thinner fibers with high crystallinity and high amounts of *β* phase should enhance the piezoelectric activity of fibrous PVDF. In order to decrease fibers’ diameters by using a surfactant, the tailoring of its chemical and physical properties to the PVDF functionality is necessary. The prepared PVDF based fibrous structure is a prospective material for the pressure active sensors where the pressure is applied in the transversal direction.

## Figures and Tables

**Figure 1 nanomaterials-10-01221-f001:**
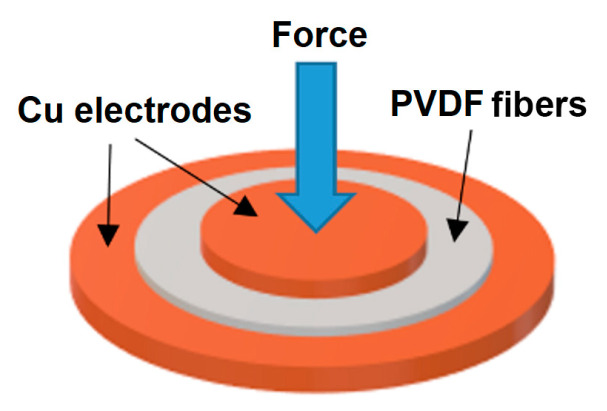
Illustrative figure of the evaluation of the piezoelectric charge coefficient in the transversal direction (*d_Tr_*) on the PVDF sample.

**Figure 2 nanomaterials-10-01221-f002:**
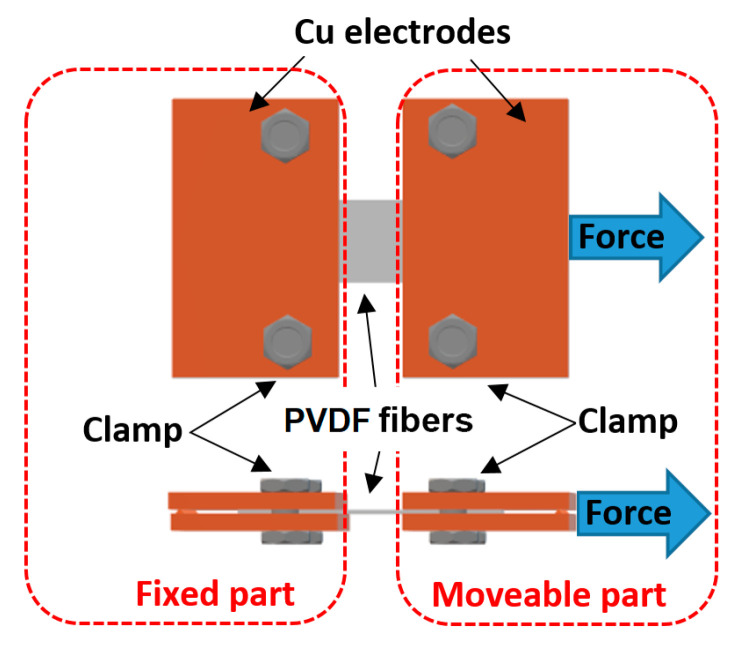
Illustrative figure of the evaluation of the piezoelectric charge coefficient in the longitudinal direction (*d_L_*_o_) on the PVDF sample.

**Figure 3 nanomaterials-10-01221-f003:**
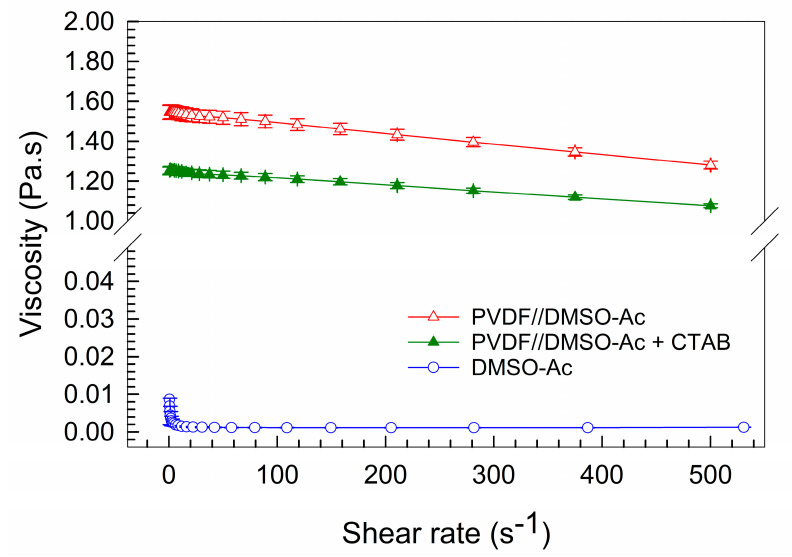
Viscosity of PVDF and solvent solutions as dependent on the shear rate. The error bars show 95% confidence intervals.

**Figure 4 nanomaterials-10-01221-f004:**
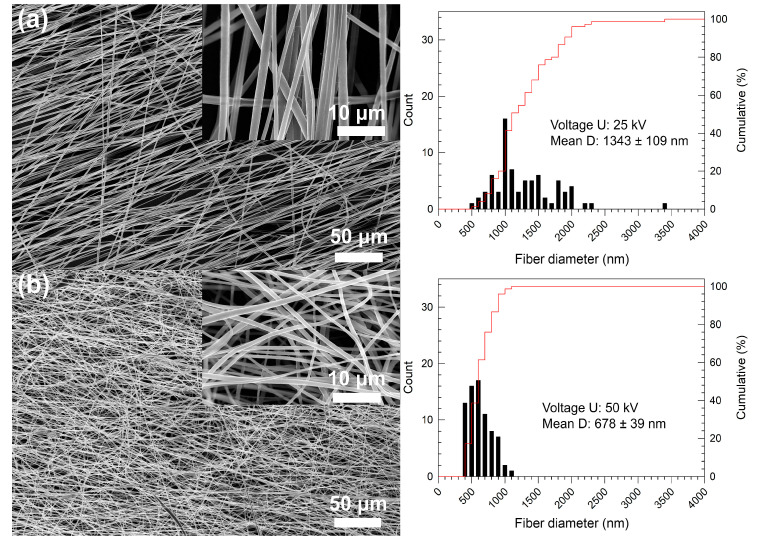
SEM images of fiber morphology and diameter distributions obtained by electrospinning at (**a**) 25 kV and (**b**) 50 kV, at a distance of 20 cm and spinning time of 15 min. The insets depict a more detailed view of the fibers’ morphology.

**Figure 5 nanomaterials-10-01221-f005:**
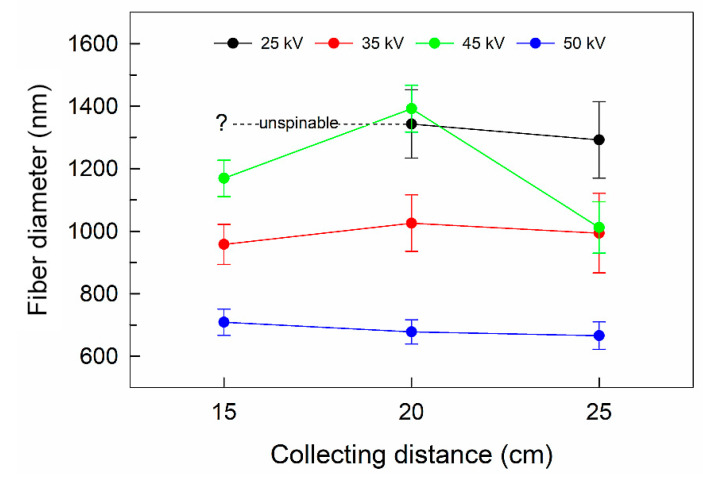
Dependence of the collecting distance on PVDF fiber diameters at different applied voltages. The error bars show 95% confidence intervals.

**Figure 6 nanomaterials-10-01221-f006:**
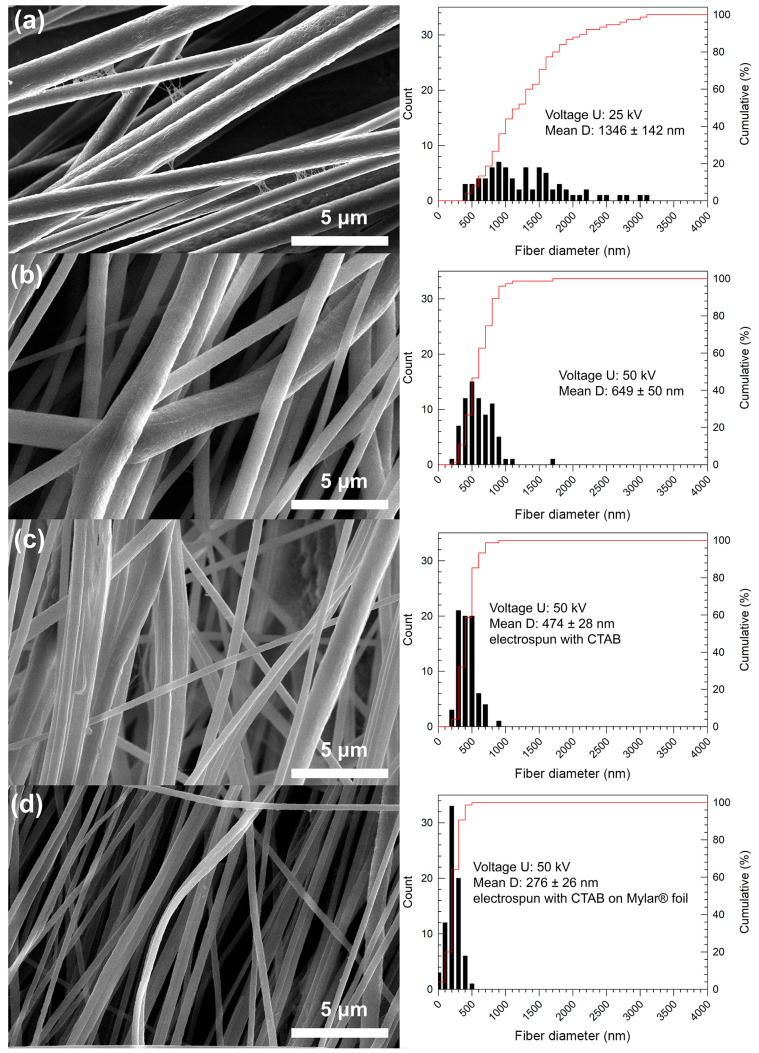
SEM images of fiber morphology and diameter distributions obtained by electrospinning of PVDF solution at (**a**) 25 kV and (**b**) 50 kV and by electrospinning of PVDF solution with CTAB at (**c**) 50 kV on aluminum foil and (**d**) 50 kV on Mylar^®^ foil at distance of 20 cm and spinning time of 90 min.

**Figure 7 nanomaterials-10-01221-f007:**
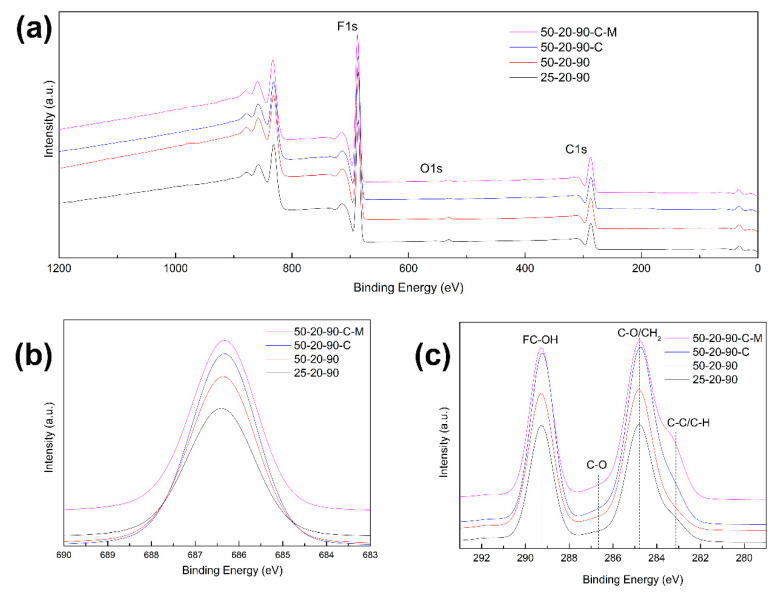
(**a**) XPS survey spectra of PVDF samples and XPS element spectra of PVDF samples for elements (**b**) F1s, (**c**) C1s and (**d**–**g**) O1s for individual PVDF samples.

**Figure 8 nanomaterials-10-01221-f008:**
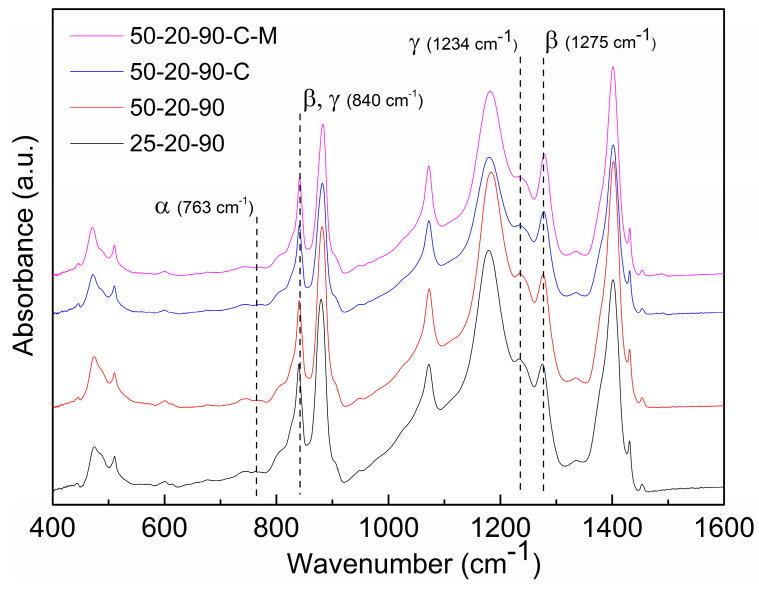
FTIR spectra of the PVDF samples.

**Figure 9 nanomaterials-10-01221-f009:**
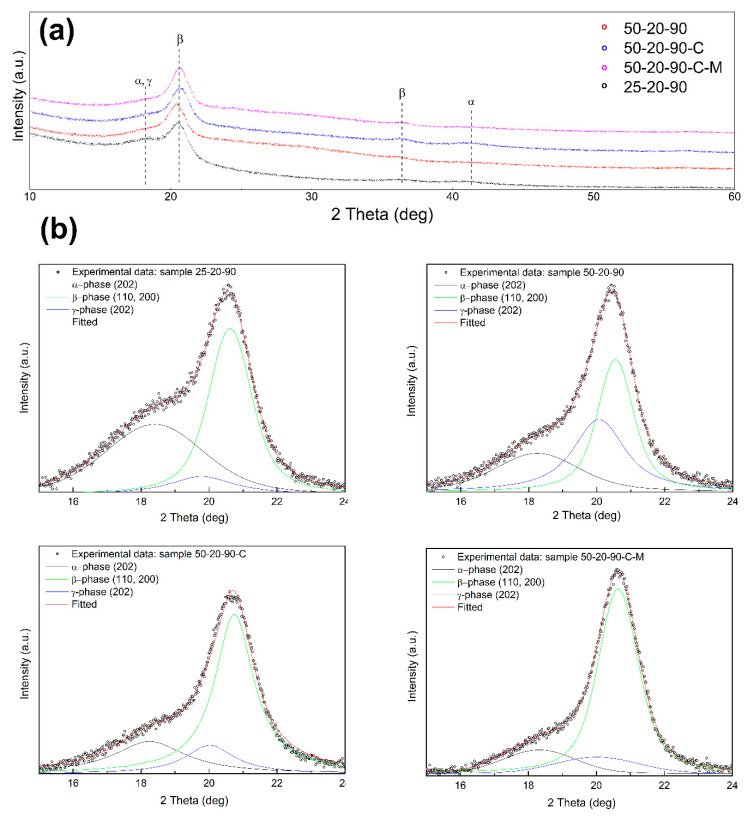
XRD patterns to define crystal phases: (**a**) XRD patterns of the PVDF samples; (**b**) individual peaks of α, β and γ crystalline phases of the PVDF samples.

**Figure 10 nanomaterials-10-01221-f010:**
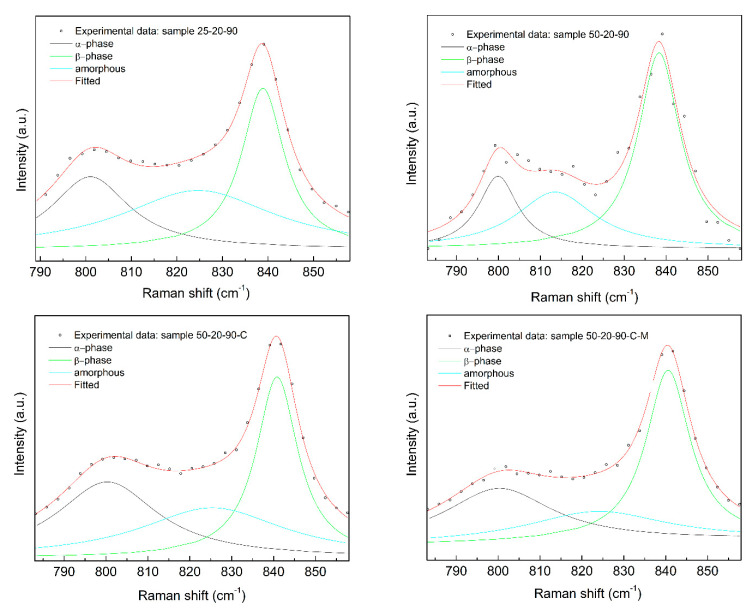
Raman spectra of the PVDF samples.

**Figure 11 nanomaterials-10-01221-f011:**
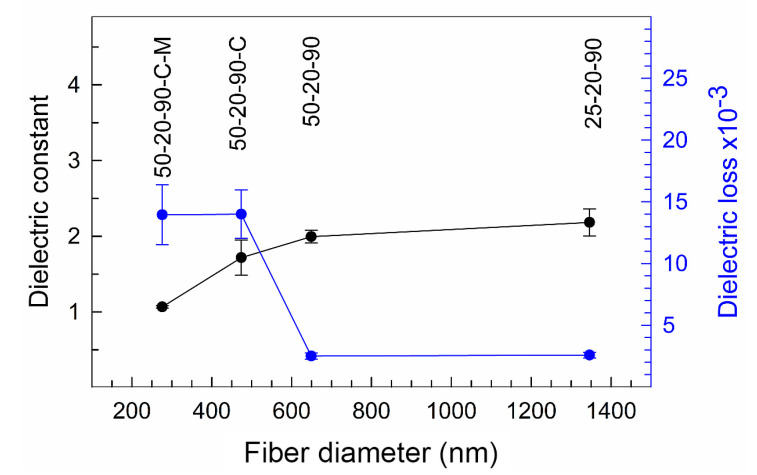
Dielectric constant and dielectric loss measured on the PVDF fibers with different diameters. The error bars show 95% confidence intervals.

**Figure 12 nanomaterials-10-01221-f012:**
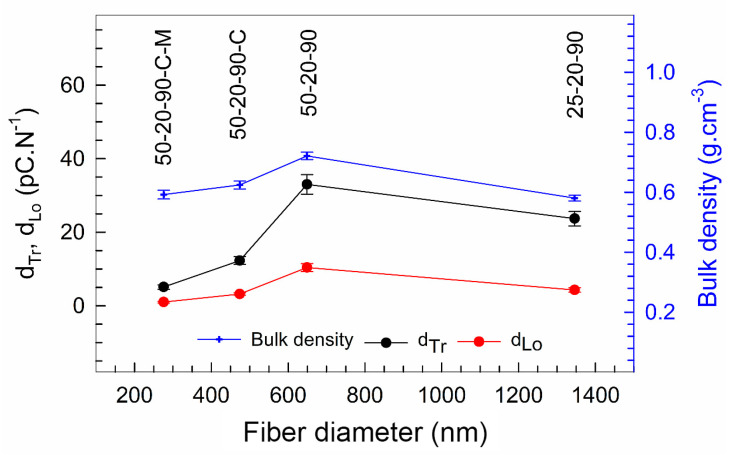
Piezoelectric charge coefficient of the PVDF fibers in the transversal (d_Tr_) and longitudinal (d_Lo_) directions and the bulk density of the PVDF samples. The error bars show 95% confidence intervals.

**Table 1 nanomaterials-10-01221-t001:** Sample abbreviations and spinning process parameters with or without ability to spin.

Sample	Voltage (kV)	Collecting Distance (cm)	Collection Time (min)	CTAB in Solution (wt%)	Collector Foil	Feeding Rate (µL min^−1^)	Comment to Process
10-15-15	10	15	15	0	aluminum	18	unspinnable
10-20-15		20					unspinnable
10-25-15		25					unspinnable
25-15-15	25	15	15	0	aluminum	18	unstable spinning
25-20-15		20					spinnable
25-25-15		25					spinnable
35-15-15	35	15	15	0	aluminum	18	spinnable
35-20-15		20					spinnable
35-25-15		25					spinnable
45-15-15	45	15	15	0	aluminum	18	spinnable
45-20-15		20					spinnable
45-25-15		25					spinnable
50-15-15	50	15	15	0	aluminum	18	spinnable
50-20-15		20					spinnable
50-25-15		25					spinnable
25-20-90	25	20	90	0	aluminum	18	spinnable
50-20-90	50	20	90	0	aluminum	18	spinnable
50-20-90-C	50	20	90	1	aluminum	20	spinnable
50-20-90-C-M	50	20	90	1	Mylar^®^	20	spinnable

**Table 2 nanomaterials-10-01221-t002:** Solution and cast film characteristics.

Sample	Conductivity (μS cm^−1^)	Viscosity ^a^ (Pa s)	Surface Tension ^b^ (mN m^−1^)	N ^c^	Crystallinity Xc (%)
DMSO-Ac	2.2	0.0086	29.9 (0.05)	3	-------
PVDF//DMSO-Ac	1.4	1.54 ± 0.03	36.6 (0.23)	3	60.6
PVDF//DMSO-Ac + CTAB	536.0	1.25 ± 0.02	35.5 (0.20)	3	80.6

^a^ Viscosity is shown for 0.3 s^−1^ shear rate and temperature of 30 °C with 95% confidence interval. ^b^ Surface tension is shown with standard deviation. ^c^ Number of measurements for conductivity, viscosity and surface tension analyses.

**Table 3 nanomaterials-10-01221-t003:** Fiber diameter and phase composition depending on process parameters of samples collected on aluminum foil (collection time of 15 min).

Sample	Voltage (kV)	Collecting Distance (cm)	Comment to Process	Fiber Diameter ^a^ (nm)	N ^b^	γ- + β-/β-Phase Content (wt%)
10-15-15	10	15	unspinnable	---	---	---
10-20-15		20	unspinnable	---	---	---
10-25-15		25	unspinnable	---	---	---
25-15-15	25	15	unstable spinning	---	---	---
25-20-15		20		1343 ± 109	75	83.2/76.2
25-25-15		25		1292 ± 122	75	86.7/79.4
35-15-15	35	15		958 ± 64	75	90.1/83.5
35-20-15		20		1026 ± 90	75	84.8/80.5
35-25-15		25		994 ± 127	75	84.2/78.9
45-15-15	45	15		1169 ± 58	75	87.2/76.9
45-20-15		20		1392 ± 75	75	90.8/78.8
45-25-15		25		1012 ± 82	75	82.0/74.8
50-15-15	50	15		709 ± 42	75	91.8/89.5
50-20-15		20		678 ± 39	75	90.3/85.7
50-25-15		25		666 ± 44	75	86.6/82.5

^a^ Fiber diameter is shown with 95% confidence interval. ^b^ Number of each individually measured fiber from SEM micrographs.

**Table 4 nanomaterials-10-01221-t004:** Characterization of the fibers electrospun for 90 min at a collecting distance of 20 cm.

Sample ^a^	Fiber Diameter ^b^ (nm)	γ- + β-/β-/γ Phase Content (wt%)	Crystallinity Xc (%)	Bulk Density ^b^ (g·cm^−3^)	Dielectric Constant ε ^b^ (-)	Dielectric Loss tanδ (10^−3^) ^b^ (-)	d_Tr_ ^b^ (pC N^−1^)	d_Lo_ ^b^ (pC N^−1^)
25-20-90	1346 ± 141	83.2/74.4/8.8	54.7	0.581 ± 0.010	2.2 ± 0.2	2.6 ± 0.2	23.7 ± 2.0	4.3 ± 0.6
50-20-90	649 ± 50	92.1/82.3/9.8	56.3	0.721 ± 0.012	2.0 ± 0.1	2.5 ± 0.2	33.0 ± 2.6	10.4 ± 1.1
50-20-90-C	474 ± 28	88.4/84.8/3.6	61.1	0.624 ± 0.013	1.7 ± 0.2	14.0 ± 2.0	12.3 ± 1.1	3.2 ± 0.3
50-20-90-C-M	276 ± 26	90.8/89.5/1.3	64.0	0.592 ± 0.015	1.1 ± 0.02	14.0 ± 2.4	5.1 ± 0.6	1.0 ± 0.2

^a^ Definitions of the sample abbreviations are given in Table 1. ^b^ Fiber diameter, bulk density, dielectric constant, dielectric loss tanδ and piezoelectric charge coefficients d_Tr_ and d_Lo_ are shown with 95% confidence interval.

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
