# Peer review of "Structure–Properties Relationship of Electrospun PVDF Fibers"

_nanomaterials, 2020, doi:10.3390/nano10061221_

Round 1

Reviewer 1 Report

This paper clearly presents the fabrication and complete characterization of the electrospun electroactive PVDF fibers materials. Electrospinning conditions (voltage collecting distance, time, adding of surfactant in the prepared solution for electrospinning) are varied and their effect on fibers morphology, crystallinity, dielectric and piezoelectric properties are studied. Figures and tables allow the reader to compare the different electrospun fibres charactteristics and the prepared solution. However, comparison with properties of other literature works to makes it possible to see where this work is positioned (bulk PVDF, other electrospun fibers should be done ). Perspectives of this work are not clearly put in evidence in the conclusions

Consequently   I would recommend publication of this work based on followings recommendations and remarks:

  1. Tables:
  • In table 1 and table 2, even if explanations related to the sample denomination “50-20-90” are found in the text, I suggest to add these explanations in the table’s captions. In table 2 caption, precise on which foil it has been collected. I would alos add in both caption the feeding rate which can have impact
  • The position of table 1 and table 2 might also be moved in results discussion part to help the reader. Number of samples or measurements could also be given in the caption, even if ii appears in the text.
  • What about the effect of the feeding rate? Was it investigated?
  • Bulk density was estimated once? Is there a  mean value for this measurement?
  1. Abbreviations related to piezoelectric or electrical can be found: Piezo activity, :Electro performance, piezo electrically. To my opinion, piezoelectric or electrical should be preferred

  1.   Line 99, referring to the sentence “the fibres were collected in the form of non-woven mats” . Could you give more informations on this step? The collection is especially important for the further piezoelectric and dielectric measurements, or the application. What about the thickness of the sample, important data to deduce for instance permitivity, or for a given application?
  2. Page 5: Could you give more details on the permittivity measurement, how many samples, what is the surface and thickness of the sample? What kind of electrodes were used?

5  Page 5, please revise the formula. C is usually the unit of the charge Q.

Indicate the unit of piezoelectric charge coefficient d in the formula or on line 155

Also, the formula doesn’t look so clear for the piezoelectric coefficient. I suggest to replace  dTr, dTo, by di where i is Tr or LO.

  1. Line , you refer to 3 orders of magnitude (0,0086 to 1,54) ; I would say 2 orders of magnitude.
  2. Line 279, it is a subtitle, shouldn’t it be in italic?
  3. Figure 11: I suggest to adjust the scale; also, as dielectric constant and dielectric loss don’t have any unit, the (-) should be removed, it is unsual for such graphic.
  4. Comparison with properties of other literature works is suggested, so that the reader can evaluate the progress on these results (bulk PVDF, other electrospun fibers)
  5. The conclusion could be completed with some perspectives of this work

Reviewer 2 Report

This study investigates the different structure-properties of PVDF fibers synthesized by electrospinning process. Also, this study showed the optimized electrospinning process for preparing a certain (micro, nano) size of fiber. Considering that this study would be very valuable to develop a novel PVDF fiber, it is noteworthy that the author made a good effort to analyze and test the materials at various conditions.

However, it is difficult to understand the objective of the study since the author did not clearly mention it. The reviewer thinks the study just showed a simple application using synthesized PVDF fiber in this study.

Comments:

  1. The sentence of the abstract and conclusion section is very similar.
  2. It’s difficult to recognize the objective of the study. The author should suggest how the synthesized material can be used efficiently.
  3. The results of the relationship between collecting distance and fiber diameter are required more discussion (why the collecting distance does not significantly affect the fiber diameter?)

Reviewer 3 Report

The manuscript presents a detailed study related to the electrospinning method for producing poly(vinylidene fluoride) (PVDF) nanofibres.
Influence of all related parameters and chemical and physical parameters
of solutions on the fibres morphology, crystallinity, phase composition,
dielectric and piezoelectric characteristics are studied and presented well. 

The manuscript can enrich readers interested in electrospinning method.

It can be accepted for publication.
